# Continuous Control with Coarse-to-fine Reinforcement Learning

**Younggyo Seo**[*]    **Jafar Uruç**    **Stephen James**

Dyson Robot Learning Lab

**Abstract:** Despite recent advances in improving the sample-efficiency of reinforcement learning (RL) algorithms, designing an RL algorithm that can be practically deployed in real-world environments remains a challenge. In this paper, we present Coarse-to-fine Reinforcement Learning (CRL), a framework that trains RL agents to *zoom-into* a continuous action space in a *coarse-to-fine* manner, enabling the use of stable, sample-efficient value-based RL algorithms for fine-grained continuous control tasks. Our key idea is to train agents that output actions by iterating the procedure of (i) discretizing the continuous action space into multiple intervals and (ii) selecting the interval with the highest Q-value to further discretize at the next level. We then introduce a concrete, value-based algorithm within the CRL framework called Coarse-to-fine Q-Network (CQN). Our experiments demonstrate that CQN significantly outperforms RL and behavior cloning baselines on 20 sparsely-rewarded RLBench manipulation tasks with a modest number of environment interactions and expert demonstrations. We also show that CQN robustly learns to solve real-world manipulation tasks within a few minutes of online training. Project website: `younggyo.me/cqn`.

**Keywords:** Reinforcement Learning, Sample-Efficient, Action Discretization

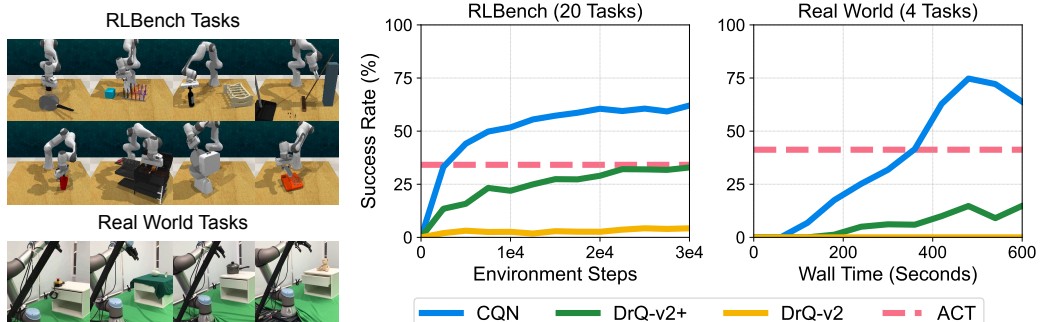

Figure 1: **Summary of results.** In sparsely-rewarded visual robotic manipulation tasks from RLBench [1] and real-world environments, CQN learns to solve the tasks with a modest number of environment interactions and expert demonstrations, outperforming baselines such as DrQ-v2 [2], its highly optimized variant DrQ-v2+, and ACT [3]. Real-world RL videos are available at our webpage.

## 1   Introduction

Recent reinforcement learning (RL) algorithms have made significant advances in learning end-to-end continuous control policies from online experiences [4, 5, 6, 7, 8, 9]. However, these algorithms often require a large number of online samples for learning robotic skills [6, 9], making it impractical for real-world environments where practitioners need to deal with resetting procedures and hardware failures. Therefore, recent successful approaches in learning visuomotor policies for real-world tasks

---

[*]Correspondence to `mail@younggyo.me`

8th Conference on Robot Learning (CoRL 2024), Munich, Germany.

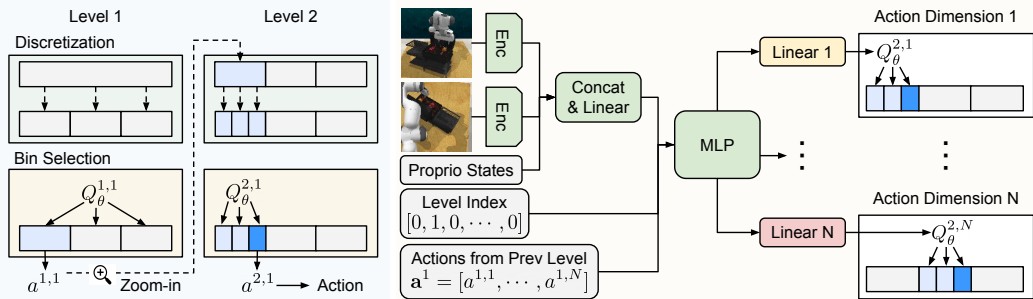

| (a) Coarse-to-fine inference procedure | (b) Coarse-to-fine critic architecture |

Figure 2: **Coarse-to-fine reinforcement learning.** (a) We design our RL agent to zoom-into the continuous action space in a *coarse-to-fine* manner by repeating the procedure of (i) discretizing the continuous action space into multiple intervals and (ii) selecting the interval with the highest Q-value to further discretize at the next level. We then use the centroid of the last level's interval as an action. (b) Our coarse-to-fine critic architecture takes input features along with one-hot level indices and actions from the previous level, and then outputs Q-values for different action dimensions. This design enables the critic to know the current level and which part of the continuous action space to zoom-into.

have mostly been methods that learn from static offline datasets, such as offline RL [10] or behavior cloning (BC) [3, 11, 12, 13]. But these offline approaches are inherently limited because they cannot improve through online experiences and thus their performance is constrained by offline data.

In this paper, we argue that many challenges in applying RL to continuous control domains arise from using actor-critic algorithms [4, 14], which introduce a separate actor network and use it for updating a critic network. Despite recent advances in stabilizing actor-critic algorithms [2, 7, 15, 16], they often suffer from instabilities due to the complex interactions between actor and critic networks [17, 18]. In contrast, value-based RL algorithms are conceptually simpler and more stable, as they operate solely with a critic, yet have achieved remarkable successes in various domains [19, 20, 21, 22]. However, value-based RL algorithms are inherently designed for use in environments with discrete actions. To exploit the benefits of value-based RL algorithms in continuous control domains, recent efforts have focused on enabling their use by discretizing the continuous action space into multiple intervals [23, 24, 25, 26]. However, this discretization scheme encounters a trade-off between the precision of actions and sample-efficiency: while more intervals are needed for fine-grained robotic tasks [10], an increased number of actions can make RL training and exploration be more difficult [25, 26, 27].

**Contribution**    To enable the use of value-based RL algorithms for fine-grained continuous control tasks without such a trade-off, we present Coarse-to-fine Reinforcement Learning (CRL), a framework that trains RL agents to *zoom-into* the continuous action space in a *coarse-to-fine* manner. Our key idea is to train an agent that outputs actions by repeating the procedure of (i) discretizing the continuous action space into multiple intervals and (ii) selecting the interval with the highest Q-value to further discretize at the next level (see Figure 2a). Unlike prior single-level approaches that need a large number of bins for high-precision [23, 25], our framework enables fine-grained control with as few as 3 bins per level (see Figure 3). Within this new CRL framework, we introduce Coarse-to-fine Q-Network (CQN), a value-based RL algorithm for continuous control (see Figure 2b), and demonstrate that it robustly learns to solve a range of continuous control tasks in a sample-efficient manner.

In particular, through extensive experiments in a demo-driven RL setup with access to a modest number of environment interactions and expert demonstrations, we demonstrate that CQN robustly learns to solve a variety of sparsely-rewarded visual robotic manipulation tasks from RLBench [1] and real-world environments. Our results are intriguing because our experiments do not use pre-training, motion planning, keypoint extraction, camera calibration, depth, and hand-designed rewards. Moreover, we show that CQN is generic and applicable to diverse benchmarks other than visual robotic manipulation; we demonstrate that CQN achieves competitive performance to actor-critic RL baselines [2, 7] in widely-used robotic tasks from DMC [28] environment with shaped rewards.

## 2 Related Work

**Actor-critic RL algorithms for continuous control** Most prior applications of RL to continuous control have been based on actor-critic algorithms [2, 4, 5, 7, 15, 16, 29, 30, 31, 32, 33, 34] that introduce a separate, parameterized actor network as a policy [14]. This is because they allow for addressing one of the main challenges in applying Q-learning to continuous domains, *i.e.,* finding continuous actions that maximize Q-values. However, in continuous control domains, actor-critic algorithms are known to be brittle and often suffer from instabilities due to the complex interactions between actor and critic networks [17, 18], despite recent efforts to stabilize them [7, 15, 16]. To address this limitation, several approaches proposed to discretize the continuous action space and learn discrete policies for continuous control. For instance, Tang and Agrawal [35] learned a policy in a factorized action space and Seyde et al. [36] learned a bang-bang controller with actor-critic RL algorithms. This paper introduces a framework that enables the use of both actor-critic and value-based RL algorithms for learning discrete policies that can solve fine-grained control tasks.

**Value-based RL algorithms for continuous control** Despite their simple critic-only architecture, value-based RL algorithms have achieved remarkable successes [19, 20, 21, 22]. However, because they require a discrete action space, there have been recent efforts to enable their use for continuous control by applying discretization to a continuous action space [10, 23, 26, 24, 25, 37] or by learning high-level discrete actions from offline data [38, 39]. For instance, some works have proposed training an autoregressive critic by treating each action dimension as a separate action to avoid the curse of dimensionality from action discretization [10, 37]. Our work is orthogonal to this, as our coarse-to-fine approach can be combined with this idea. On the other hand, several works have demonstrated that training factorized critics for each action dimension can achieve competitive performance to actor-critic algorithms [24, 25]. However, this single-level discretization may not be scalable to domains requiring high-precision actions, as such domains typically necessitate fine-grained discretization [10]. To address this limitation, Seyde et al. [26] proposed gradually enlarging action spaces throughout training, but this introduces a challenge of constrained optimization. In contrast, our CRL framework enables us to learn discrete policies for continuous control in a stable and simple manner.

Notably, the closest work to ours is C2F-ARM [40] that trains value-based RL agents to zoom-into a voxelized 3D robot workspace by predicting the voxel to further discretize. C2F-ARM is a special case of our CRL framework, where the agent operates as a hierarchical, next-best pose agent [34]; it splits the robot manipulation problem into high-level next-best-pose control and low-level control (usually a motion planning) problems. CQN on the other hand, is more general and can be used for any action mode, including joint control. We provide additional discussion in Appendix F.

## 3 Method

We present Coarse-to-fine Reinforcement Learning (CRL), a framework that trains RL agents to *zoom-into* a continuous action space in a *coarse-to-fine* manner (see Section 3.1). Within this framework, we introduce Coarse-to-fine Q-Network (CQN), a value-based RL algorithm for continuous control (see Section 3.2) and describe various design choices for improving CQN in visual robotic manipulation tasks (see Section 3.3). We provide the overview and pseudocode in Figure 2 and Appendix B.

### 3.1 Framework: Coarse-to-fine Reinforcement Learning

To enable the use of value-based RL algorithms for learning discrete policies in fine-grained continuous control domains, we propose to formulate the continuous control problem as a multi-level discrete control problem via *coarse-to-fine* action discretization. Specifically, given a number of levels $L$ and a number of bins $B$, we apply discretization to the continuous action space $L$ times (see Figure 3), in contrast to prior approaches that discretize action space into multiple intervals in a single-level [25, 41]. We then train RL agents to *zoom-into* the continuous action space by repeating the procedure of (i) discretizing the continuous action space at the current level into $B$ intervals and (ii) selecting the interval with the highest Q-value to further discretize at the next level (see Figure 2a).

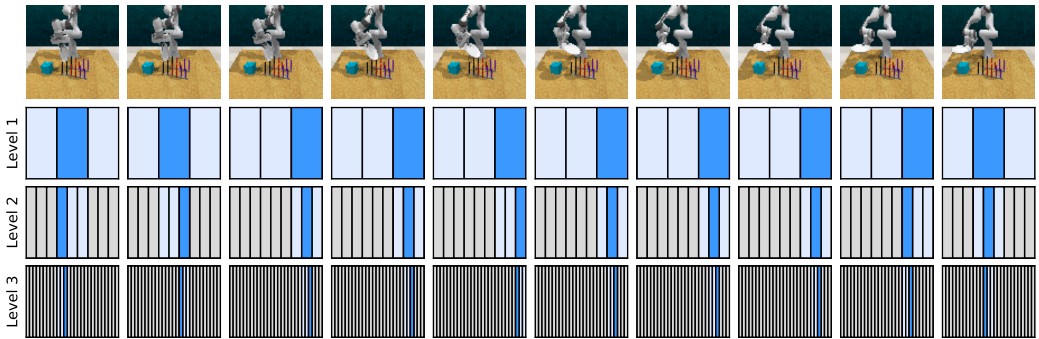

Figure 3: **Examples of coarse-to-fine discretization.** With a pre-defined number of levels ($L$) and intervals ($B$), *e.g.,* $L = 3$ and $B = 3$ in this example, we apply discretization to the continuous action space $L$ times with different precisions. We then design our RL agents to learn a critic network with only a few actions at each level, *e.g.,* 3 actions in this example, conditioned on previous level's actions. This enables us to learn discrete policies that can output high-precision actions while avoiding the difficulty of learning the critic network with a large number of discrete actions.

Our intuition is that, by designing our agents to learn a critic network with only a few discrete actions at each level (*i.e.,* $B$ actions), our coarse-to-fine framework can effectively allow for learning discrete policies that can output high-precision actions while avoiding the difficulty of learning the critic network with a large number of discrete actions (*e.g.,* $B^L$ actions is required for achieving the same precision with a single-level discretization). Here we note that our framework is compatible with both actor-critic and value-based RL algorithms as they can operate with discrete actions. But this paper focuses on developing a value-based RL algorithm because of its simple and stable critic-only architecture (see Section 3.2), and leaves the development of actor-critic RL algorithm as future work.

## 3.2 Algorithm: Coarse-to-fine Q-Network

**Problem setup** We formulate a vision-based continuous control problem as a partially observable Markov decision process [42, 43], where, at each time step $t$, an agent encounters an observation $\mathbf{o}_t$, selects an action $\mathbf{a}_t$, receives a reward $r_{t+1}$, and encounters a new observation $\mathbf{o}_{t+1}$ from an environment. Our goal is to learn a policy that maximizes the expected sum of rewards through RL in a sample-efficient manner, *i.e.,* by using as few online samples as possible.

**Inputs and encoder** We consider an observation $\mathbf{o}_t$ consisting of pixel observations $(\mathbf{o}_t^{v_1}, ..., \mathbf{o}_t^{v_V})$ captured from viewpoints $(v_1, ..., v_V)$ and low-dimensional proprioceptive states $\mathbf{o}_t^{\texttt{low}}$. We then use a lightweight 4-layer convolutional neural network (CNN) encoder $f_\theta^{\texttt{enc}}$ to encode pixels $\mathbf{o}_t^{v_i}$ into visual features $\mathbf{h}_t^{v_i}$, *i.e.,* $\mathbf{h}_t^{v_i} = f_\theta^{\texttt{enc}}(\mathbf{o}_t^{v_i})$. To fuse information from view-wise features, we concatenate features from all viewpoints and project them into low-dimensional features. Then we concatenate fused features with proprioceptive states $\mathbf{o}_t^{\texttt{low}}$ to construct features $\mathbf{h}_t$.

**Coarse-to-fine critic architecture** Let $a_t^{l,n}$ be an action at level $l$ and action dimension $n$ (*e.g.,* delta angle for $n$-th joint of a robotic arm) and $\mathbf{a}_t^l = (a_t^{l,1}, ..., a_t^{l,N})$ be an action at level $l$ where $\mathbf{a}_t^0$ is defined as a zero action vector. By following the design of Seyde et al. [25] that introduce factorized Q-networks for different action dimensions, we define our coarse-to-fine critic to consist of individual Q-networks at level $l$ and action dimension $n$ as below (see Figure 2b for an illustration):

$$Q_\theta^{l,n}(\mathbf{h}_t, a_t^{l,n}, \mathbf{a}_t^{l-1}) \text{ for } n \in \{1, ..., N\} \text{ and } l \in \{1, ..., L\} \tag{1}$$

We note that our design mainly differs from prior work with a single-level critic [24, 25] in that our Q-network takes $\mathbf{a}_t^{l-1}$, *i.e.,* actions from all dimensions at previous level, to enable each Q-network to be aware of other networks' decisions at the previous level. We also design our critic to share most of parameters for all levels and dimensions by sharing linear layers except the last linear layer [41] and making Q-networks take one-hot level index as inputs[2].

---

[2]We omit one-hot level index from the equation for the simplicity of notation.

**Inference procedure**   We describe our coarse-to-fine inference procedure for selecting actions at time step $t$ (see Figure 2a and Appendix B for the illustration and pseudocode of our inference procedure). We first introduce constants $a_t^{n,\texttt{low}}$ and $a_t^{n,\texttt{high}}$ that are initialized with $-1$ and $1$ for each action dimension $n$. For all action dimensions $n$, we repeat the following steps for $l \in \{1, ..., L\}$:

- Step 1 (Discretization): We discretize an interval $[a_t^{n,\texttt{low}}, a_t^{n,\texttt{high}}]$ into $B$ uniform intervals, each of which becomes the action space for Q-network $Q_\theta^{l,n}$.

- Step 2 (Bin selection): We find $\text{argmax}_{a'} Q_\theta^{l,n}(\mathbf{h}_t, a', \mathbf{a}_t^{l-1})$ for each $n$, which corresponds to the interval with the largest Q-value. We then set $a_t^{l,n}$ to the centroid of the selected interval and concatenate actions from all dimensions into $\mathbf{a}_t^l$.

- Step 3 (Zoom-in): We set $a_t^{n,\texttt{low}}$ and $a_t^{n,\texttt{high}}$ to the minimum and maximum value of the selected interval, zooming into the selected intervals within the action space.

We use the last level's action $\mathbf{a}_t^L$ as the action at time step $t$. In practice, we parallelize the procedures across all the action dimensions $n$ for faster inference. We further describe a procedure for computing Q-values with input actions, along with its pseudocode, in Appendix B.

**Q-learning objective**   Q-learning objective for action dimension $n$ at level $l$ is defined as below:

$$\mathcal{L}_{\text{RL}}^{l,n} = \left(Q_\theta^{l,n}(\mathbf{h}_t, a_t^{l,n}, \mathbf{a}_t^{l-1}) - r_{t+1} - \gamma \max_{a'} Q_{\bar{\theta}}^{l,n}(\mathbf{h}_{t+1}, a', \pi^{l-1}(\mathbf{h}_{t+1}))\right)^2 \tag{2}$$

where $\bar{\theta}$ are delayed critic parameters updated with Polyak averaging [44] and $\pi^l$ is a policy that outputs the action $\mathbf{a}_t^l$ at each level $l$ via the inference steps with our critic, *i.e.,* $\pi^l(\mathbf{h}_t) = \mathbf{a}_t^l$.

**Implementation and training details**   We use the 2-layer dueling network [45] and a distributional critic [46] with 51 atoms. By following Hafner et al. [47], we use layer normalization [48] with SiLU activation [49] for every linear and convolutional layers. We use AdamW optimizer [50] with weight decay of 0.1 by following Schwarzer et al. [51]. Following prior work that learn from offline data [52, 53], we sample minibatches of size 256 each from the online replay buffer and the demonstration replay buffer, resulting in a total batch size of 512. More details are available in Appendix C.

### 3.3   Optimizations for Visual Robotic Manipulation

We describe various design choices for improving CQN in visual robotic manipulation tasks.

**Auxiliary behavior cloning objective**   Following the idea of prior work [54, 55], we introduce an auxiliary behavior cloning (BC) objective that encourages agents to imitate expert actions. Specifically, given an expert action $\tilde{\mathbf{a}}_t$, we introduce an auxiliary margin loss [56] that encourages $Q(\mathbf{h}_t, \tilde{\mathbf{a}}_t^l)$ to be higher than Q-values of non-expert actions $Q(\mathbf{h}_t, \mathbf{a}_t^l)$ for all levels $l$ as below:

$$\mathcal{L}_{\text{BC}}^{l,n} = \max_{a'} \left(Q_\theta^{l,n}(\mathbf{h}_t, a', \mathbf{a}_t^{l-1}) + f^{\texttt{margin}}(\tilde{a}_t^{l,n}, a')\right) - Q_\theta^{l,n}(\mathbf{h}_t, \tilde{a}_t^{l,n}, \tilde{\mathbf{a}}_t^{l-1}) \tag{3}$$

where $f^{\texttt{margin}}$ is a function that gives 0 when $a' = \tilde{a}_t^{l,n}$ and a margin value $m$ otherwise. This objective encourages Q-values for expert actions to be at least higher than other Q-values by $m$. We describe how we modify BC objective to align better with the distributional critic in Appendix A.

**Relabeling successful online trajectories as demonstrations**   Inspired by the idea of self-imitation learning [57] that encourages agents to reproduce their own good decisions, we label the successful trajectories from environment interaction as demonstrations. We find that this simple scheme can be helpful for RL training by widening the distribution of demonstrations throughout training.

**Environment interaction**   Similar to prior value-based RL algorithms [51, 58], we choose actions using the target Q-network to improve the stability throughout environment rollouts. Moreover, as we find that standard exploration techniques of injecting noises [4, 59, 60] make it difficult to solve fine-grained control tasks, we instead add a small Gaussian noise with standard deviation of 0.01.

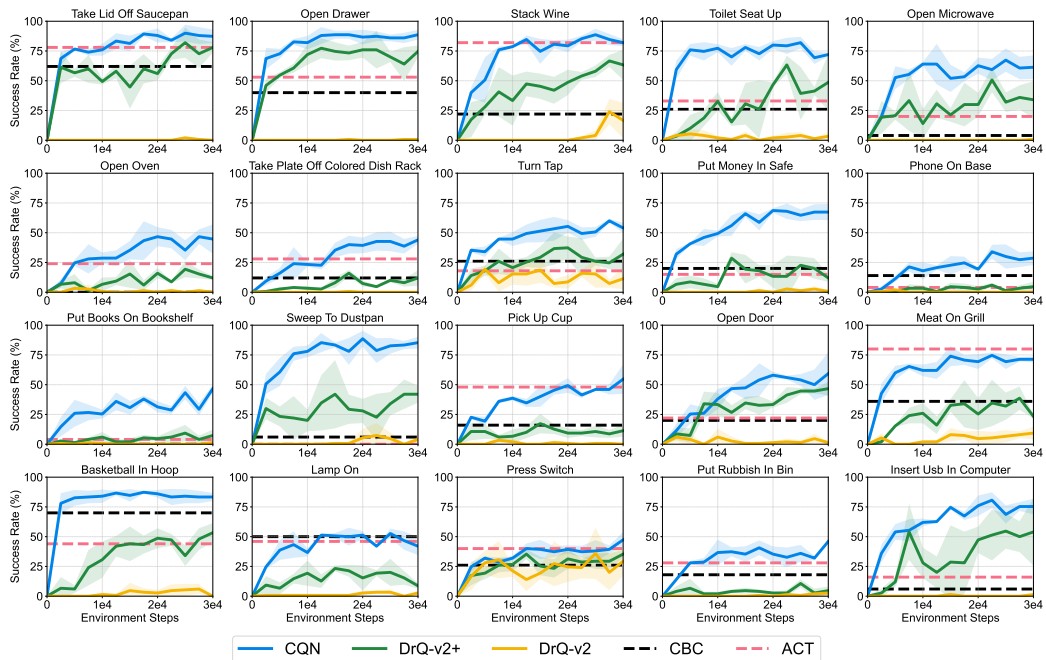

Figure 4: **Simulation results** on 20 sparsely-rewarded tasks from RLBench [1]. All experiments are initialized with 100 expert demonstrations and all RL methods have an auxiliary BC objective. We report the success rate over 25 episodes. The solid line and shaded regions represent the mean and confidence intervals, respectively, across 3 runs.

## 4 Experiments

We design our experiments to investigate the following questions: (i) How does CQN compare to previous RL and BC baselines? (ii) Can CQN be sample-efficient enough to be practically used in real-world environments? (iii) How do various design factors of CQN affect the performance?

### 4.1 RLBench Experiments

**Setup** For quantitative evaluation, we mainly consider a demo-driven RL setup where we aim to solve visual robotic manipulation tasks from RLBench [1] environment with access to a limited number of environment interactions and expert demonstrations[3]. Unlike prior work that designed experiments to make RLBench tasks less challenging by using hand-designed rewards [55, 61] or heuristics that depend on motion planning, *e.g.,* keypoint extraction [34, 40], we consider a sparse-reward setup without the use of motion planner. Specifically, we label the reward of the last timestep in successful episodes as $1.0$ and train RL agents to output the difference of joint angles at each time step. We use RGB observations with $84 \times 84$ resolution captured from front, wrist, left-shoulder, and right-shoulder cameras. Proprioceptive states consist of 7-dimensional joint positions and a binary gripper state. For all tasks, we use the same set of hyperparameters, *e.g.,* 3 levels and 5 bins, without tuning them for each task. See Appendix C for more details.

**RL baselines** Because CQN is a generic value-based RL algorithm compatible with other techniques for improving value-based RL [51, 58] or demo-driven RL [52, 53, 62, 63], we mainly focus on comparing CQN against representative baselines to which comparison can highlight the benefit of our framework. To this end, we first consider DrQ-v2 [2], a widely-used actor-critic RL algorithm, as our RL baseline. Moreover, for a fair comparison, we design our strong RL baseline: DrQ-v2+, a highly optimized variant of DrQ-v2 that incorporates a distributional critic and our recipes for manipulation tasks (see Section 3.3). We also note that all RL methods have an auxiliary BC objective.

---

[3]We provide experimental results in state- and vision-based robotic tasks from DMC [28] in Appendix E.

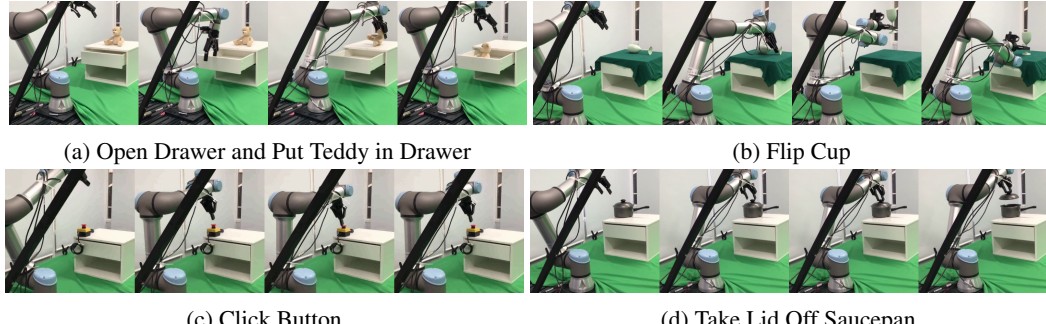

(a) Open Drawer and Put Teddy in Drawer  (b) Flip Cup

(c) Click Button  (d) Take Lid Off Saucepan

Figure 5: **Real-world tasks** used in our real-world experiments (see Appendix D for more details).

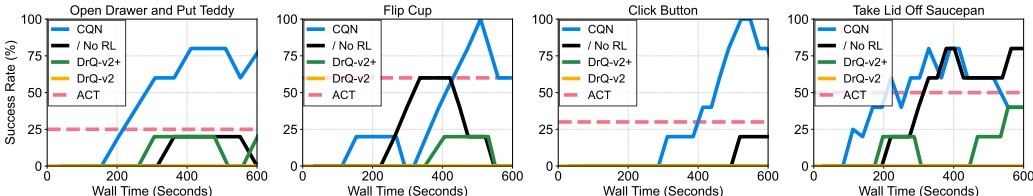

Figure 6: **Real-world results.** Learning curves on 4 real-world manipulation tasks, measured by the success rate. We run experiments for 10 minutes and report the running mean across 5 episodes.

**BC baselines**    To demonstrate the benefit of learning through online experiences, we consider ACT [3], which learns to predict a sequence of actions, as our BC baseline. We choose ACT because it achieves competitive performance to other methods such as DiffusionPolicy [11]. We also consider an additional BC baseline, *i.e.,* Coarse-to-fine BC (CBC), which shares every detail with CQN such as action discretization and architecture but trained only with BC objective.

**Results**    In Figure 4, we find that CQN consistently outperforms actor-critic RL baselines, *i.e.,* DrQ-v2 and DrQ-v2, in terms of both sample-efficiency and asymptotic performance. In particular, CQN significantly outperforms our highly-optimized baseline DrQ-v2+ by a large margin, highlighting the benefit of our CRL framework that allows the use of value-based RL algorithm for continuous control. Moreover, we observe that CQN can quickly match the performance of BC baselines (*i.e.,* ACT and CBC) and surpass them in most of the tasks, highlighting the benefit of learning by trial and error.

### 4.2    Real-world Experiments

**Setup**    We further demonstrate the effectiveness of CQN in real-world tasks that use a UR5 robot arm with 20 to 50 human-collected demonstrations (see Figure 5 for examples of real-world tasks). Unlike RLBench experiments that take one update step per every environment step, we take 50 or 100 update steps between episodes to avoid jerky motions during the environment interaction. All RL methods have an auxiliary BC objective and we report the running mean across 5 recent episodes. For ACT, we report the average success rate over 20 episodes to evaluate it with the same randomization range used in RL experiments. We use stack of 4 observations as inputs and 4 levels with 3 bins. Unless otherwise specified, we use the same hyperparameters as in RLBench experiments for all methods, which shows the robustness of CQN to hyperparameters. See Appendix D for more details.

**Results**    In Figure 6, we observe intriguing results where CQN can learn to solve complex real-world tasks within 10 minutes of online training, while a baseline without RL objective often fails to do so. In particular, we find that this baseline without RL objective nearly succeeds in solving the task but makes a mistake in states that require high-precision actions, which demonstrates the benefit of RL similar to the results in simulated RLBench environment (see Table 1c). Moreover, we observe that the training of DrQ-v2+ is unstable especially when it encounters unseen observations during training. In contrast, CQN robustly learns to solve the tasks and consistently outperforms DrQ-v2+ in all tasks. We provide full videos of real-world RL training for all tasks in our project website.

| Level | Bin | SR |
|---|---|---|
| 1 | 5 | 8.8% |
| 1 | 17 | 30.7% |
| 1 | 65 | 51.2% |
| 1 | 256 | 39.5% |
| 3 | 5 | **77.5%** |
| 3 | 17 | 65.5% |

(a) Bins

| Level | SR |
|---|---|
| 1 | 8.8% |
| 2 | 55.8% |
| 3 | **77.5%** |
| 4 | 72.8% |
| 5 | 46.5% |
| 6 | 37.8% |

(b) Levels

| $\mathcal{L}_{RL}$ | $\mathcal{L}_{BC}$ | C51 | SR |
|---|---|---|---|
| ✗ | ✓ | - | 36.5% |
| ✓ | ✗ | ✓ | 1.8% |
| ✓ | ✓ | ✗ | 16.7% |
| ✓ | ✓ | ✓ | **77.5%** |

(c) Objectives

| Action Selection | Expl. Noise | SR |
|---|---|---|
| Online | $\mathcal{N}(0, 0.01)$ | 70.2% |
| Target | ✗ | 75.1% |
| Target | $\mathcal{N}(0, 0.1)$ | 50.8% |
| Target | $\mathcal{N}(0, 0.01)$ | **77.5%** |

(d) Exploration

Table 1: **Analysis and ablation studies.** We investigate the effect of (a) bins and (b) levels. (c) We investigate the effect of RL objective ($\mathcal{L}_{RL}$), BC objective ($\mathcal{L}_{BC}$), and the use of distributional critic (C51) [46]. (d) We investigate the effect of using target Q-network for action selection and small exploration noise. SR denotes success rate and default settings are highlighted in gray .

### 4.3 Analysis and Ablation Studies

We investigate the effect of hyperparameters and various design choices by running experiments on 4 tasks from RLBench. We provide more analysis and ablation studies in Appendix A.

**Effect of levels and bins** In Table 1a and Table 1b, we investigate the effect of levels and bins within CQN. As shown in Table 1a, we find that single-level baseline performance peaks at 65 bins and decreases after it, which shows the limitation of single-level action discretization that struggles to scale up to tasks that require high-precision actions. Moreover, we find that 3-level CQN also struggles with more bins, as learning Q-networks with more actions can be difficult. In Table 1b, we find that 3 or 4 levels are sufficient and performance keeps decreasing with more levels. We hypothesize this is because learning signals from levels with too fine-grained actions may confuse the network with limited capacity because of sharing parameters for all the levels.

**Effect of objectives and distributional critic** In Table 1c, we investigate the effect of RL and BC objectives, along with the effect of using distributional critic (*i.e.,* C51) [46]. To summarize, we find that (i) RL objective is crucial as in real-world experiments (see Section 4.2), (ii) auxiliary BC objective is crucial as RL agents struggle to keep close to demonstration distribution without the BC loss, and (iii) distributional critic is important; severe value overestimation makes RL training unstable in the initial phase of RL training without the distributional critic.

**Effect of exploration** We further investigate the effect of how our agents do exploration, *i.e.,* which network to use for selecting actions and how to add noise to actions, in Table 1d. We find that using target Q-network for selecting actions outperforms using online Q-network. We hypothesize this is because (i) Polyak averaging [44] can improve the generalization [64] and (ii) online network changes throughout episode. We also find that using a small Gaussian noise with $\mathcal{N}(0, 0.01)$ outperforms a variant with a strong noise because manipulation tasks require high-precision actions.

## 5 Discussion

We present CRL, a framework that enables the use of value-based RL algorithms in fine-grained continuous control domains, and CQN, a concrete value-based RL within this framework. Our key idea is to train RL agents to zoom-into a continuous action space in a coarse-to-fine manner. Extensive experiments demonstrate that CQN efficiently learns to solve a range of continuous control tasks.

**Limitations and future directions** Overall, we are excited about the potential of our framework and there are many exciting future directions: supporting high update-to-data ratio [51, 58, 65], 3D representations [55, 66, 67, 68, 69, 70, 71, 72], tree-based search [20, 73], and bootstrapping RL from BC [62, 74] or offline RL [75, 76, 77], to name but a few. One particular limitation we are keen to address is that we still need quite a number of demonstrations. Reducing the number of demonstrations by incorporating pre-trained models [78, 79, 80] or augmentation techniques [81, 82, 83] would be an interesting future direction. We discuss more limitations and future directions in Appendix G.

## Acknowledgements

Big thanks to the members of the Dyson Robot Learning Lab for discussions and infrastructure help: Nic Backshall, Nikita Chernyadev, Iain Haughton, Richie Lo, Yunfan Lu, Xiao Ma, Sumit Patidar, Sridhar Sola, Mohit Shridhar, Eugene Teoh, and Vitalis Vosylius.

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

| $\mathcal{L}_{\text{C51-BC}}$ | Relabeling | Centralized critic | SR |
|---|---|---|---|
| ✗ | ✓ | ✓ | 72.3% |
| ✓ | ✗ | ✓ | 57.8% |
| ✓ | ✓ | ✗ | 76.3% |
| ✓ | ✓ | ✓ | **77.5%** |

(a) Effect of design choices and optimizations

| Action mode | Scaling | SR |
|---|---|---|
| Absolute | ✓ | 20.5% |
| Delta | ✗ | 71.5% |
| Delta | ✓ | **77.5%** |

(b) Action mode and scaling

| Stack | SR |
|---|---|
| 1 | 63.7% |
| 2 | 75.0% |
| 4 | 76.0% |
| 8 | **77.5%** |

(c) History

Table 2: **Additional analysis and ablation studies.** We investigate the effect of BC objective for C51 ($\mathcal{L}_{\text{C51-BC}}$), relabeling successful episodes as demonstrations, and using centralized critic [25]. We also investigate the effect of (b) action mode and scaling and (c) using a history of observations. SR denotes success rate and default settings are highlighted in  gray .

# A    Additional Analysis and Ablation Studies

Here, we provide additional analysis and ablation studies in Table 2. For results in this section and Section 4, we report aggregate results on 4 tasks: Turn Tap, Stack Wine, Open Drawer, Sweep To Dustpan, with 3 runs for each task.

**Auxiliary BC with distributional critic**    We find that our BC objective in Equation 3 is often not synergistic with distributional critic, because it leads to a shortcut of increasing Q-values (*i.e.,* the mean of value distribution) by increasing the probability mass of atoms corresponding to supports with large values. To address this issue, given an expert action $\tilde{a}_t$, we introduce a BC objective that encourages a distribution with the expert action $Q(s, \tilde{a}_t)$ to be preferred over $Q(s, a_t)$ instead of only using the mean of the distribution as a metric.

Our idea is to utilize the concept of first-order stochastic dominance [84, 85]: when a random variable $A$ is first-order stochastic dominant over a random variable $B$, for all outcome $x$, $F_A(x) \leq F_B(x)$ holds, with strict inequality at some x. Intuitively, this means that $A$ is preferred over $B$ because the $A$ is more likely to have a higher outcome $x$. Based on this, we design an auxiliary BC objective that encourages $Q(s, \tilde{a}_t)$ to be stochastically dominant over $Q(s, a_t)$, *i.e.,* $\mathcal{L}_{\text{C51-BC}}$, which encourages RL agents to prefer the distribution induced by expert actions $\tilde{a}_t$ to non-expert actions $a_t$. In Table 2a, we find that using $\mathcal{L}_{\text{C51-BC}}$ achieves 77.5%, outperforming a variant that uses $\mathcal{L}_{\text{BC}}$ that achieves 72.3%.

**Centralized critic**    Our coarse-to-fine critic architecture is based on the design of Seyde et al. [25] that train a factorized critic across action dimensions. However, we do not use the centralized critic training scheme as in the original paper, because (i) we find that using the average Q-value as an objective is not aligned well with the use of distributional critic and (ii) our design can already facilitate critics for different dimensions to share information as they are conditioned on actions from the previous level (see Figure 2b). Indeed, as shown in Table 2a, we find that using such an objective does not make a significant difference in performance; thus we do not use it for simplicity.

**Relabeling successful episodes as demonstrations**    We investigate the effectiveness of our relabeling scheme (see Section 3.3) in Table 2a, where we observe that performance largely drops without the scheme. Though this is effective in our RLBench experiments, we note that this idea depends on the characteristic of our manipulation tasks where successful episodes can be treated as optimal trajectories; investigating the effectiveness of it with noisy offline data or suboptimal demonstrations can be an interesting direction.

**Action mode**    We investigate how the choice of action mode between the absolute joint control or delta joint control affects the performance. We find that using the delta joint action mode significantly outperforms a baseline with the absolute action mode. We hypothesize this is because delta joint control's action space is narrower and makes it easy to learn fine-grained control policies. Moreover, we observe that using the absolute joint action mode in real-world environments often leads to dangerous behaviors and robot failures in practice because of large movements between each step.

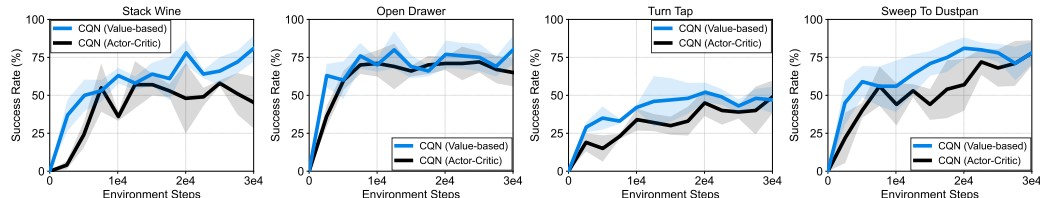

Figure 7: **Simulation results** on 20 sparsely-rewarded tasks from RLBench [1]. All experiments are initialized with 100 expert demonstrations and all RL methods have an auxiliary BC objective. We report the success rate over 25 episodes. The solid line and shaded regions represent the mean and confidence intervals, respectively, across 3 runs.

**Data-driven action scaling** For all experiments, we follow James and Davison [34] that compute the minimum and maximum actions from the demonstrations and scale actions using these values as the action space bounds. We investigate the effect of this scaling scheme in Table 2b, where we find that this makes it easy to learn to solve manipulation tasks.

**Using a history of observations** Similar to prior researches that show the effectiveness of using a history of observations when training IL agents for robotic manipulation [11, 86], we find that using stacked observations [19] is also crucial when training RL agents for manipulation in Table 2c.

**Comparison to CQN with actor-critic baseline** To further support our claims on the effectiveness of value-based RL for continuous control, in Figure 7, we provide experimental results with an actor-critic baseline that trains the critic in the exact same way but learns a separate parameterized policy to maximize the value function. We find CQN (Value-based) achieves better sample-efficiency compared to actor-critic baseline, showing the benefit of utilizing a simple critic-only algorithm. Nonetheless, having a separate actor has benefits such as better action inference scheme, robust to Q-value scale for exploration, and others, which is an interesting direction we are keen to explore.

# B  Pseudocode

In this section, we first provide an inference procedure for computing Q-values. We then provide the pseudocode of inference procedures and CQN training in Algorithm 1 and Algorithm 2.

**Inference procedure for computing Q-values**  We describe the procedure for computing Q-values when actions $\mathbf{a}_t$ are given as inputs, which is similar to action selection procedure in Section 3.2. We first introduce constants $a_t^{n,\texttt{low}}$ and $a_t^{n,\texttt{high}}$ that are initialized with $-1$ and $1$ for each action dimension $n$. For all action dimensions $n$, we repeat the following steps for $l \in \{1, ..., L\}$:

- Step 1 (Discretization): We discretize an interval $[a_t^{n,\texttt{low}}, a_t^{n,\texttt{high}}]$ into $B$ uniform intervals, each of which becomes the action space for Q-network $Q_\theta^{l,n}$.

- Step 2 (Bin selection): We find the interval that contains given input actions $\mathbf{a}_t$ and compute Q-value $Q_\theta^{l,n}(\mathbf{h}_t, a_t^{l,n}, \mathbf{a}_t^{l-1})$ for the selected interval.

- Step 3 (Zoom-in): We set $a_t^{n,\texttt{low}}$ and $a_t^{n,\texttt{high}}$ to the minimum and maximum value of the selected interval, zooming into the selected intervals within the action space.

We then obtain the set of Q-values $\{Q_\theta^{l,n}(\mathbf{h}_t, a_t^{l,n}, \mathbf{a}_t^{l-1})\}$.

---

**Algorithm 1** Coarse-to-fine inference procedure

---

1: **Inputs:** Features $\mathbf{h}_t$, number of levels $L$, intervals $B$, and action dimensions $N$
2: **Optional inputs:** Input actions $\mathbf{a}_t$
3: Initialize $a_t^{n,\texttt{low}}, a_t^{n,\texttt{high}}$ to -1 and 1 for all $n$
4: Initialize $\mathbf{a}_t^0$ to $\mathbf{0}$
5: **for** each level $l \in (1, ..., L)$ **do**
6:   **for** each dimension $n \in (1, ..., N)$ **do**
7:     // STEP 1: DISCRETIZATION
8:     Discretize an interval $[a_t^{n,\texttt{low}}, a_t^{n,\texttt{high}}]$ to $B$ intervals

9:     // STEP 2: BIN SELECTION
10:    **if** Input actions $\mathbf{a}_t$ are given **then**
11:      Find interval that contains $\mathbf{a}_t$ at the current level $l$ and dimension $n$
12:      Set $a_t^{l,n}$ as the centroid of the selected interval
13:      Compute Q-value $Q_\theta^{l,n}(\mathbf{h}_t, a_t^{l,n}, \mathbf{a}_t^{l-1})$
14:    **else**
15:      Find interval that satisfies: $\operatorname{argmax}_{a'} Q_\theta^{l,n}(\mathbf{h}_t, a', \mathbf{a}_t^{l-1})$
16:      Set $a_t^{l,n}$ as the centroid of the selected interval

17:    // STEP 3: ZOOM-IN
18:    Set $a_t^{n,\texttt{low}}, a_t^{n,\texttt{high}}$ to minimum and maximum of the selected interval
19:   **if not** Input actions $\mathbf{a}_t$ are given **then**
20:     Aggregate actions as $\mathbf{a}_t^l = (a_t^{l,1}, ..., a_t^{l,N})$
21: **if** Input actions $\mathbf{a}_t$ are given **then**
22:   **return**  Q-values $\{Q_\theta^{l,n}(\mathbf{h}_t, a_t^{l,n}, \mathbf{a}_t^{l-1})\}$ for all $l$ and $n$
23: **else**
24:   **return**  Action from the last level $\mathbf{a}_t^L$

---

---

**Algorithm 2** Coarse-to-fine Q-Network (CQN)

---

1: **Inputs:** Number of levels $L$, intervals $B$, and action dimensions $N$
2: Initialize CQN parameters $\theta$ and target parameters $\bar{\theta}$
3: Initialize a buffer $\mathcal{B}$ and a demonstration replay buffer $\mathcal{B}^{\mathsf{e}}$
4: **for** each timestep $t$ **do**
5:     // ENVIRONMENT INTERACTION
6:     Compute feature $\mathbf{h}_t$ from $\mathbf{o}_t$
7:     Get action $\mathbf{a}_t$ with **Algorithm 1**
8:     Apply $\mathbf{a}_t$ to environment and observe $\mathbf{o}_{t+1}, r_{t+1}$
9:     Add transition $(\mathbf{o}_t, \mathbf{a}_t, r_{t+1}, \mathbf{o}_{t+1})$ to replay buffer $\mathcal{B}$
10:     // UPDATE Q-NETWORK
11:     Initialize $\mathcal{L}_{\mathtt{CQN}}$ to 0
12:     Sample minibatches from $\mathcal{B}$ and $\mathcal{B}^{\mathsf{e}}$
13:     **for** for each level $l \in (1, ..., L)$ **do**
14:         **for** for each dimension $n \in (1, ..., N)$ **do**
15:             Compute $\mathcal{L}_{\mathtt{RL}}^{l,n}$ as in Equation 2 with **Algorithm 1** and samples from the minibatches
16:             Compute $\mathcal{L}_{\mathtt{BC}}^{l,n}$ as in Equation 3 with **Algorithm 1** and samples from the minibatches
17:             Update $\mathcal{L}_{\mathtt{CQN}} = \mathcal{L}_{\mathtt{CQN}} + (\lambda_{\mathtt{RL}} \cdot \mathcal{L}_{\mathtt{RL}}^{l,n} + \lambda_{\mathtt{BC}} \cdot \mathcal{L}_{\mathtt{BC}}^{l,n})/(N \cdot L)$
18:     Update $\theta$ by minimizing $\mathcal{L}_{\mathtt{CQN}}$
19:     Update $\bar{\theta} = (1 - \tau) \cdot \bar{\theta} + \tau \cdot \theta$

---

## C   Experimental Details: Simulation

**Simulation and tasks**   We use RLBench [1] simulator based on CoppeliaSim [87] and PyRep [88]. We run experiments in 20 sparsely-rewarded visual manipulation tasks with a 7-DoF Franka Panda robot arm and a parallel gripper (see Table 3 for the list of tasks).

Table 3: **RLBench tasks** with their maximum episode length used in our experiments.

| Task | Length | Task | Length |
|---|---|---|---|
| Take Lid Off Saucepan | 100 | Put Books On Bookshelf | 175 |
| Open Drawer | 100 | Sweep To Dustpan | 100 |
| Stack Wine | 150 | Pick Up Cup | 100 |
| Toilet Seat Up | 150 | Open Door | 125 |
| Open Microwave | 125 | Meat On Grill | 150 |
| Open Oven | 225 | Basketball In Hoop | 125 |
| Take Plate Off Colored Dish Rack | 150 | Lamp On | 100 |
| Turn Tap | 125 | Press Switch | 100 |
| Put Money In Safe | 150 | Put Rubbish In Bin | 150 |
| Phone on Base | 175 | Insert Usb In Computer | 100 |

**Data collection**   For demonstration collection, we modify the maximum velocity of a Franka Panda robot arm by 2 times in PyRep, which shortens the length of demonstrations without largely degrading the quality of demonstrations. We use RLBench's dataset generator for collecting 100 demonstrations.

**Computing hardware**   For all RLBench experiments, we use a single 72W NVIDIA L4 GPU with 24GB VRAM and it takes 6.5 hours for training both CQN and DrQ-v2+. We find that major bottleneck is slow simulation because our model consists of lightweight CNN and MLP architectures.

**Hyperparameters**   We use the same set of hyperparameters for all the RLBench tasks. We provide detailed hyperparameters of CQN in Table 4 and DrQ-v2/DrQ-v2+ in Table 5.

# D    Experimental Details: Real-world

**Tasks**    We design 4 real-world visual robotic manipulation tasks with different characteristics. We do not provide partial reward during the episode and only provide reward 1 at the end of fully successful episode. See Figure 8 for pictures that show how we randomize the initial position of the objects between each episode. We describe the tasks in more detail as below:

- **Open Drawer and Put Teddy in Drawer.** The goal of this task is to (i) fully open the drawer, which is slightly open at the start of each episode, (ii) pick up the teddy bear, and (iii) put the teddy bear in the drawer. We use 50 demonstrations for this task. We randomize the initial position of the teddy bear between every episode in a 10cm radius circle.

- **Flip Cup.** The goal of this task is to (i) grasp the handle of a plastic wine glass and (ii) flip the cup in a upright position. We use 20 demonstrations for this task. We randomize the initial position of the cup between every episode in a 15×30cm rectangular region.

- **Click Button.** The goal of this task is to click the button with the closed gripper. We use 21 demonstrations for this task. We randomize the initial position of the button between every episode in a 38×38cm squared region.

- **Take Lid Off Saucepan.** The goal of this task is to (i) grasp the lid of the saucepan and (ii) lift the lid up. We use 24 demonstrations for this task. We randomize the initial position of the saucepan between every episode in a 38×38cm squared region.

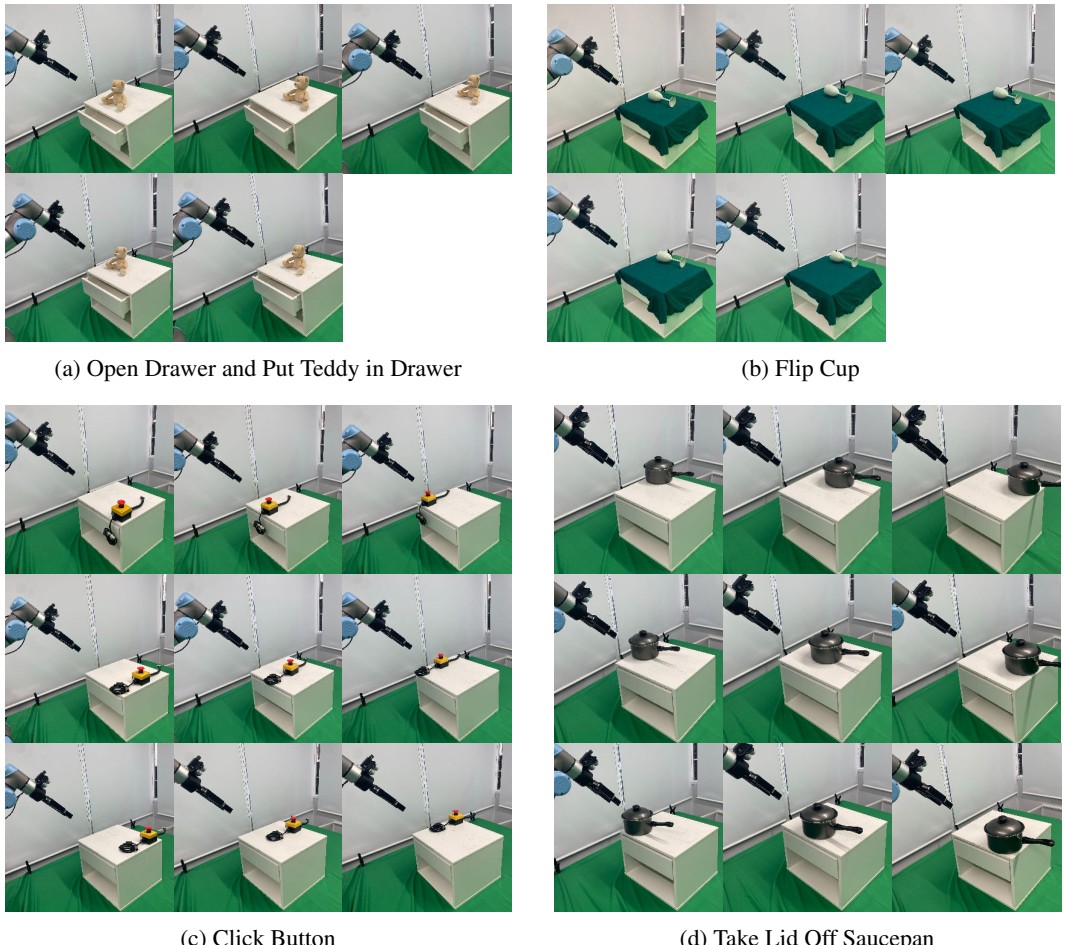

(a) Open Drawer and Put Teddy in Drawer

(b) Flip Cup

(c) Click Button

(d) Take Lid Off Saucepan

Figure 8: **Randomization for real-world tasks.** We provide pictures that show how we randomize the initial position of the objects in our real-world experiments.

**Robot and computing hardware**    We use a 6-DoF UR5e robot arm with a Robotiq-2F-140 gripper for our real-world experiments. For cameras, we use left-shoulder, right-shoulder, upper-wrist, lower-wrist RealSense D435 cameras, without camera calibration and depth, to capture RGB observations with $640 \times 480 \times 3$ resolution. We use a single 230W NVIDIA RTX A5500 GPU with 24GB VRAM. Each action inference takes 0.008s in average, thus our model operates at $\sim$125Hz in execution time.

**Data collection**    We use teleoperation with a joint mirroring system, where a human controls a leader robot and a follower robot mirrors the movement in the joint space. We record RGB observations and 6-DoF joint positions during the demonstration collection phase, and downsize RGB pixels to $84 \times 84 \times 3$ resolution. We also preprocess demonstrations by filtering out some timesteps where the robot *pauses*, which happens when a human operator stops controlling the robot. Specifically, we remove timesteps when the difference in joint positions between between two consecutive timesteps is smaller than the pre-specified threshold. We use smaller thresholds for `Click Button` and `Take Lid Off Saucepan` as we find that preprocessing with large thresholds often removes timesteps corresponding to clicking button or grasping the lid.

**Real-world RL pipeline**    For all the tasks and methods, we train the model for 10 minutes of wall time that includes time for training models and robot execution time. We implement a human reward user interface system (see Figure 9), which supports pause/unpause of the robot, labelling the episode as success or failure, and resetting the robot failure cases. We use binary reward (*i.e.,* 1 for success and 0 for failure) for all experiments. We also do not use success detector or automated reset procedures. Instead, human practitioners label the episodes and reset the scene.

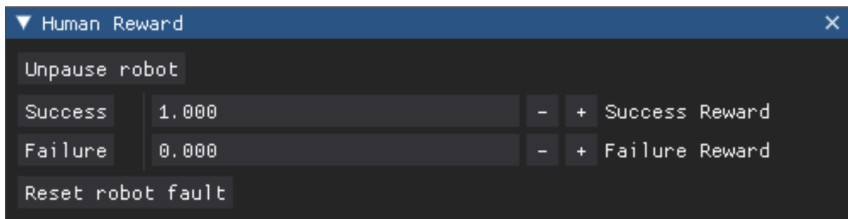

Figure 9: **Human Reward user interface** used in our real-world experiments.

**Hyperparameters**    As we previously mentioned in Section 4.2, we do episodic training where we take a fixed number of update steps between each episode. We take 100 update steps for `Open Drawer` and `Put Teddy in Drawer` task and 50 update steps for all the other tasks, as the former task is a long-horizon task compared to other tasks and thus has larger demonstration sizes. We provide detailed hyperparameters of CQN in Table 4 and DrQ-v2/DrQ-v2+ in Table 5.

Table 4: CQN hyperparameters used in RLBench and Real-world experiments.

| Hyperparameter | Value |
|---|---|
| Image resolution | $84 \times 84 \times 3$ |
| Image augmentation (RLBench) | RandomShift [2] (RLBench) |
| Image augmentation (Real-world) | RandomShift [2], Brightness, Contrast |
| Frame stack | 8 (RLBench) / 4 (Real-world) |
| CNN - Architecture | Conv (c=[32, 64, 128, 256], s=2, p=1) |
| MLP - Architecture | Linear (c=[64, 512, 512], bias=False) |
| CNN & MLP - Activation | SiLU [49] and LayerNorm [48] |
| C51 - Atoms | 51 |
| C51 - $v_{min}$, $v_{max}$ | -1, 1 |
| CQN - Levels | 3 (RLBench) / 4 (Real-world) |
| CQN - Bins | 5 (RLBench) / 3 (Real-world) |
| BC loss ($\mathcal{L}_{BC}$) scale | 1.0 |
| RL loss ($\mathcal{L}_{RL}$) scale | 0.1 |
| Relabeling as demonstrations | True |
| Data-driven action scaling | True |
| Action mode | Delta Joint |
| Exploration noise | $\epsilon \sim \mathcal{N}(0, 0.01)$ |
| Target critic update ratio ($\tau$) | 0.02 |
| N-step return | 3 |
| Training interval | Every step (RLBench) / Every episode (Real-world) |
| Training steps | 1 (RLBench) / 100 (Teddy), 50 (Otherwise) |
| Batch size | 256 |
| Demo batch size | 256 |
| Optimizer | AdamW [50] |
| Learning rate | 5e-5 |
| Weight decay | 0.1 |

Table 5: DrQ-v2 [2] and DrQ-v2+ hyperparameters used in RLBench and Real-world experiments.

| Hyperparameter | Value |
|---|---|
| Image resolution | $84 \times 84 \times 3$ |
| Image augmentation (RLBench) | RandomShift [2] |
| Image augmentation (Real-world) | RandomShift [2], Brightness, Contrast |
| Frame stack | 8 (RLBench) / 4 (Real-world) |
| CNN - Architecture | Conv (c=[32, 64, 128, 256], s=2, p=1) |
| MLP - Architecture | Linear (c=[64, 512, 512], bias=True) |
| CNN & MLP - Activation | ReLU |
| C51 - Atoms | 101 (DrQ-v2+) / Not used (DrQ-v2) |
| C51 - $v_{min}$, $v_{max}$ | -1, 1 (DrQ-v2+) / Not used (DrQ-v2) |
| BC loss ($\mathcal{L}_{BC}$) scale | 1.0 |
| RL loss ($\mathcal{L}_{RL}$) scale | 1.0 |
| Relabeling as demonstrations | True (DrQ-v2+) / False (DrQ-v2) |
| Data-driven action scaling | True (DrQ-v2+) / False (DrQ-v2) |
| Action mode | Delta joint |
| Exploration noise | $\epsilon \sim \mathcal{N}(0, 0.01)$ (DrQ-v2+) / $\epsilon \sim \mathcal{N}(0, 0.2)$ (DrQ-v2) |
| Target critic update ratio ($\tau$) | 0.01 |
| N-step return | 3 |
| Training interval | Every step (RLBench) / Every episode (Real-world) |
| Training steps | 1 (RLBench) / 100 (Teddy), 50 (Otherwise) |
| Batch size | 256 (DrQ-v2+) / 512 (DrQ-v2) |
| Demo batch size | 256 (DrQ-v2+) / 0 (DrQ-v2) |
| Optimizer | AdamW [50] |
| Learning rate | 1e-4 |
| Weight decay | 0.1 (DrQ-v2+) / 0.0 (DrQ-v2) |

# E DeepMind Control Experiments

**Setup** To demonstrate that CQN can achieve competitive performance in widely-used, shaped-rewarded RL benchmarks, we provide experimental results in a variety of continuous control tasks from DeepMind Control Suite (DMC) [28]. We also note that DMC benchmark consists of a variety of low-dimensional and high-dimensional control tasks, enabling us to evaluate the scalability of CQN on environments with high-dimensional action spaces. For baselines, we compare CQN to RL algorithms that learn continuous policies, whose performances in DMC are publicly available[45]. For state-based control tasks, we consider soft actor-critic (SAC) [7] as our baseline. For vision-based control tasks, we compare CQN to DrQ-v2 [2]. For hyperparameters, we follow the original hyperparameters used in the publicly available results. For instance, we use the action repeat of 1 for state-based control tasks and action repeat of 2 for vision-based control tasks. For CQN hyperparameters, we set minimum and maximum value bounds to 0 and 200 for distributional critic and use 3 levels with 5 intervals for coarse-to-fine action discretization.

**Results** Figure 10 and Figure 11 show that CQN achieves competitive or superior performance to RL baselines that learn continuous policies in most of the tasks. This result demonstrates that our framework is generic, *i.e.,* it can be used for state-based, vision-based, sparsely-rewarded, and densely-rewarded environments. One trend we observe in pixel-based DMC tasks is that the performance of CQN often stagnates early in locomotion tasks (*e.g.,* Quadruped, Hopper, and Walker), unlike in manipulation tasks where CQN achieves superior performance to the baseline. We hypothesize this is because we use a naïve exploration scheme: we use the exploration noise of $\epsilon \sim \mathcal{N}(0, 0.1)$. It would be an interesting future direction to investigate how to design exploration schedule that can exploit a discrete action space from our coarse-to-fine discretization scheme.

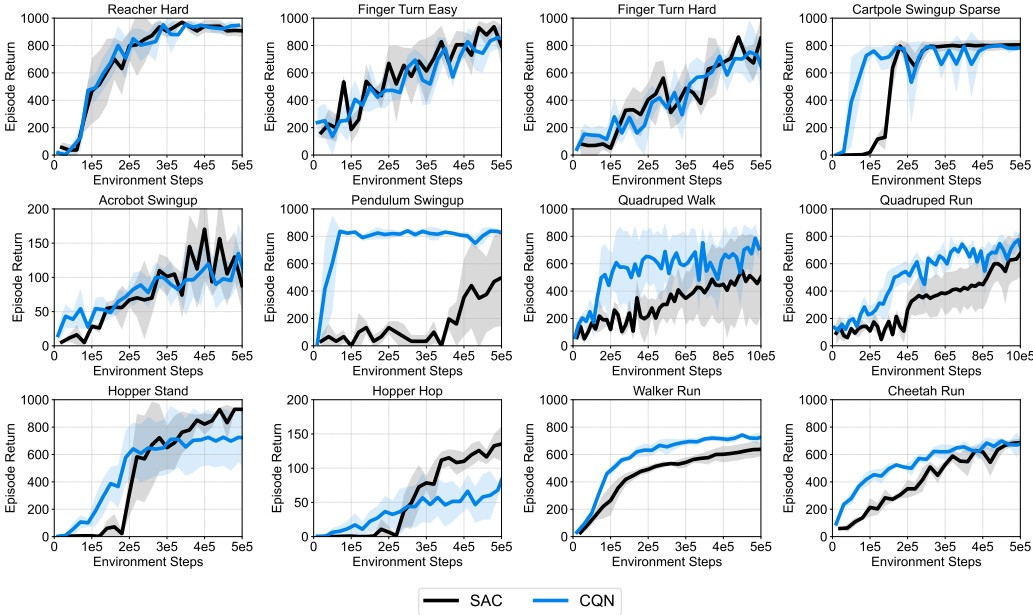

Figure 10: **State-based DMC results.** Learning curves on 12 state-based robotic locomotion tasks from DeepMind Control Suite [28], measured by the episode return. The solid line and shaded regions represent the mean and confidence intervals, respectively, across 4 runs.

[4]DrQ-v2: https://github.com/facebookresearch/drqv2/
[5]SAC:https://github.com/denisyarats/pytorch_sac

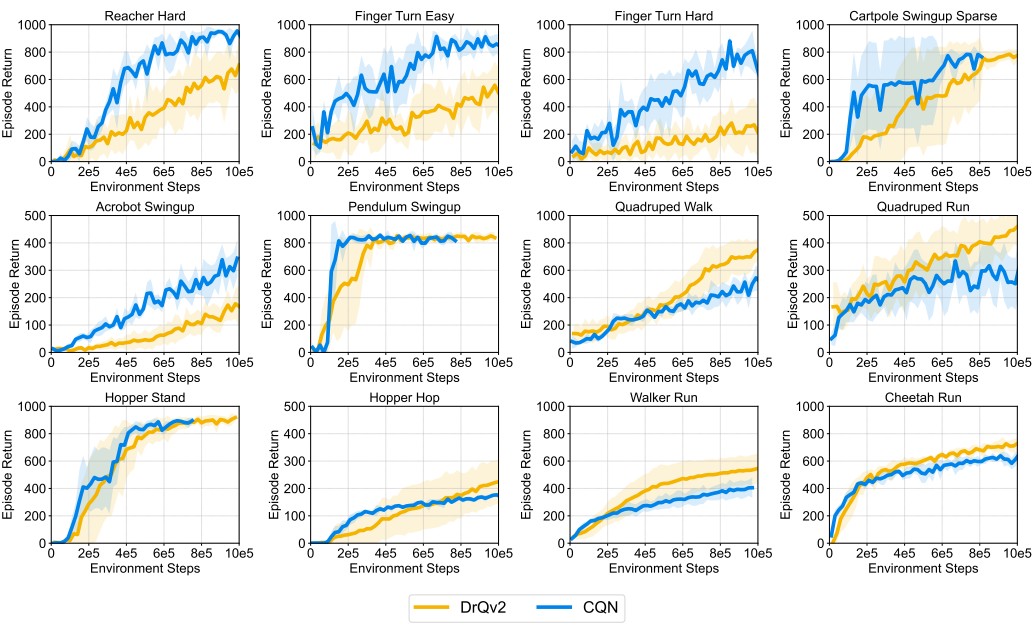

Figure 11: **Pixel-based DMC results.** Learning curves on 12 pixel-based robotic locomotion tasks from DeepMind Control Suite [28], measured by the episode return. The solid line and shaded regions represent the mean and confidence intervals, respectively, across 4 runs.

# F    Additional Related Work

**Real-world RL for continuous control**    Obviously, our work is not the first application of RL to real-world continuous control domains. In particular, in the context of learning locomotion behaviors, there have been impressive successes in demonstrating the capability of RL controllers trained in simulation and then transferred to real-world environments [89, 90, 91, 92, 93, 94]. More closely related to our work are approaches that have demonstrated RL can be used to learn robotic skills directly in real-world environments, with state inputs [95, 96, 97, 98, 99], visual inputs [29, 33, 100, 101, 102], and offline data [77, 103, 104], addressing challenges such as exploration, state estimation, camera calibration, robot failure, and the cost of resetting procedures. Moreover, there has also been a progress in developing benchmarks that can serve as a proxy for real-world experiments [1, 105] and developing a software package for easily deploying RL algorithms to real-world RL [106]. Investigating the effectiveness of our framework on such various benchmarks and real-world domains would be an exciting future direction we are keen to explore.

**Hierarchical RL**    Our work is loosely related to approaches that learn hierarchical RL agents [107, 108] that trains high-level RL agents that provides goals (options or skills) and low-level RL agents that learn to follow goals or behave conditioned on goals [109, 110, 111, 112, 113, 114]. This is because our approach also introduces a multi-level, hierarchical structure in the action space. But our work is different in that we introduce a hierarchy by splitting the fixed, general continuous action space but hierarchical RL approaches typically introduce a temporally or behaviorally abstracted action as a high-level action (goal, option, or skill). Nevertheless, it would be an interesting future direction to incorporate such abstract high-level actions into our coarse-to-fine critic architecture, as it is straightforward to condition our critic on such abstract actions by introducing an additional level.

# G  Limitations and Future Directions

**Data augmentation**   In this work, we applied very simple data augmentations: RandomShift [2] that shifts pixels by 4 pixels, brightness augmentation, and contrast augmentation. However, as shown in recent works that investigated the effectiveness of augmentations for learning visuomotor policies [115, 116], applying more strong augmentations can also be helpful for improving the generalization capability of RL agents. Moreover, applying augmentation to images with generative models [117] can further enhance the generalization capability of RL agents to unseen environments. Incorporating such strong augmentations potentially with techniques for stabilizing RL training [82, 83] can be an interesting future direction.

**Advanced vision encoder and representation learning**   CQN uses a simple, light-weight visual encoder, *i.e.,* 4-layer CNN encoder, and also a naïve way of fusing view-wise features that concatenates image features. While this has an advantage of having a simple architecture and thus a very fast inference speed, incorporating an advanced vision encoder architectures such as ResNet [118] or Vision Transformer [119] may improve the performance in tasks that require fine-grained control. Moreover, given the recent improvements in learning multi-view representations [55, 66, 120] or generating 3D models [121, 122, 123, 124, 125], incorporating such improvements and 3D prior into encoder design can be helpful for improving the sample-efficiency of CQN, especially in tasks that require multi-view information as already shown in recent several behavior cloning approaches [67, 68, 69, 70, 71, 72]. Learning such representations by pre-training the visual encoder on large multi-view datasets [126, 127, 128] would also be an interesting direction.

**Handling a history of observations**   For taking a history of observations as inputs, we follow a very simple scheme of Mnih et al. [19] that stacks observations. However, this might not be scalable to long-horizon tasks where such a stacking of 4 or 8 observations may not provide a sufficient information required for solving the target tasks. In that sense, designing a model-based RL algorithm within our CRL framework based on recent works [61, 47, 129] or incorporating architectures that can handle a sequence of observations, such as RNNs [130, 131], Transformers [132], and state-space models [133], can be a natural future direction to our work.

**Training with high update-to-data ratio**   Recent work have demonstrated the effectiveness of using high update-to-data (UTD) ratio (*i.e.,* number of update steps per every environment step) for improving the sample-efficiency of RL algorithms [51, 58, 65]. In this work, we used 1 UTD ratio in RLBench experiments for faster experimentation as using higher UTD ratio slows down training. This slow-down in training speed can be an issue in real-world experiments where practitioners often need to be physically around the robot and monitor the progress of training for labelling the episode or safety reason. Thus, investigating the performance of CQN with high UTD by utilizing a design or software that supports asynchronous training [33, 106] would be an interesting future direction we are keen to explore. Furthermore, we note that recent approaches typically depend on *resetting* technique for supporting high-UTD but such resetting can be problematic in that it may lead to dangerous behaviors with real robots. Investigating how to support high UTD without such a resetting technique can be also an interesting future direction especially in the context of real-world RL.

**Search-based action selection**   CQN uses a simple inference scheme that greedily selects an interval with the highest Q-value from the first level. However, there is a room for improvement in action selection by incorporating search algorithms that exploit the discrete action space [73].

**Bootstrapping from offline data with BC or offline RL**   While our experiments show that CQN can quickly match and outperform the performance of BC baseline such as ACT [3], there is a room for improvement by investigating how to bootstrap RL training from offline RL [75, 76, 77] or BC policies [62, 74]. For instance, pre-training CQN agents with offline RL techniques on robot learning dataset [134, 135] or utilizing a separate BC policy pre-trained on demonstrations would be interesting and straightforward future directions.

**Human-in-the-loop learning**  One critical limitation of applying RL to real-world applications is that practitioners need to be physically around the robot in most cases; otherwise it involves a huge engineering to automate resetting procedures and designing a success detection system. However, this can lead to another interesting and promising future direction of leveraging human guidance in the training pipeline in the form of human-in-the-loop learning. For instance, incorporating a DAgger-like system that provides human-guided trajectory for RL agents [136], investigating a way to utilize human-labelled reward but address the subjectivity of such human labels throughout training via preference learning [137, 138] can be interesting future directions.

**Failure case: Tasks with correlated actions**  Our work is based on the idea of Seyde et al. [25] that assumes independence between action dimensions. While Seyde et al. [25] showed that this assumption holds in many scenarios, there are cases when our algorithm fails. For instance, CQN with absolute joint action mode performs worse than CQN with delta joint action mode in Table 2b. This could be seen as a failure case because action dimensions become more correlated because some large actions at specific joints (dimensions) can affect other joints. Incorporating an idea that does not assume independence between action dimensions [10, 37] is an interesting direction to investigate.

**Increased compute cost**  Because our algorithm requires (i) multiple forward passes through critics over multiple levels and (ii) the use of distributional critic, our algorithm might incur additional compute cost than the *flat* baselines. Considering the increased sample-efficiency from adopting our idea and high cost of running robots, we believe incorporating such a cost is worth it. But it would be a practically interesting direction to reduce the computational cost of our idea.

**Failure case: Dynamic tasks**  Our work mainly assume that actions close to each other in continuous space are similar, which can be particularly useful when similar actions are roughly similar but certain fine-grained action is needed to solve the task. This is because our approach can easily learn to select the coarser bin and then focus on learning to refine the behavior. However, our framework can be difficult to be applied to highly-dynamic tasks where it is difficult to assume that similar actions will have similar values, so that discretizing action space with fixed interval can make it difficult to learn Q-values (*e.g.,* actions around the bin boundary can have very different values). Further exploration on dynamic tasks and improving CQN on such tasks would be an interesting future direction.

## H  Things that did not work

We describe the methods and techniques that did not work in our RLBench experiments when we use default hyperparameters and setups from the original work.

**Small batch RL and prioritized sampling**  We tried using small batch size [139] but find that large batch size performs better in RLBench experiments. This aligns with the original observation of Obando Ceron et al. [139] where large batch size performs better with fewer number of environment interactions. We also tried using prioritized experience replay [140] but we find that it slows down training without a significant performance gain.

**Exploration with NoisyNet**  Instead of manually setting a small Gaussian noise $\mathcal{N}(0, 0.01)$, we tried using NoisyNet [59] with varying magnitudes of initial noise scale. But we find that it perturbs action too much regardless of noise scales, making it not possible to solve the manipulation tasks.

**Learning critic with classification loss**  We tried the idea of Farebrother et al. [141] that proposed to train value functions with categorical cross-entropy loss. But we find that using a distributional critic [46] works better when value bounds are set to -1 and 1 for sparsely-rewarded tasks.

**Different distributional RL algorithms**  We tried distributional RL algorithms other than C51, *i.e.,*QR-DQN [142] and IQN [143], but find no difference between them in our experiments.

**L2 feature normalization** We tried normalizing every feature vectors to have a unit norm following Hussing et al. [144] but this significantly degraded the performance in our experiments.

**RL with action chunking** Motivated by recent BC approaches that demonstrated the effectiveness of predicting a sequence of actions (*i.e.,* action chunk) [3, 11], we also tried incorporating action chunking into RL. Specifically, we expand the action space by treating actions from multiple timesteps as a single action. But we find that this naïve approach does not work well; investigating how to incorporate such an idea into RL would be an interesting future direction.

