# OpenReview forum: "Continuous Control with Coarse-to-fine Reinforcement Learning"
_robot-learning.org/CoRL/2024/Conference — CoRL 2024_

### Official Review · Reviewer_qKY2 · 2024-07-13

**Originality:** 3
**Technical Quality:** 4
**Clarity Of Presentation:** 3
**Potential Impact:** 3
**Recommendation:** 3
**Confidence:** 4

**Review:**

**Strengths:**
Overall, this paper presents a simple framework for decomposing continuous actions into a hierarchical sequence of low-dimensional discrete actions. The authors demonstrate the efficacy of Q-learning using a simple critic network on this action space on an appropriate range of simulated benchmarks and real-world tasks. The paper is clearly presented, and while there is no shortage of hierarchical learning algorithms, this method seems to be a clever but effective formulation, at least within the scope of tasks evaluated. Proposed extensions and next steps seem generally reasonable.

**Weaknesses:**
In general, some more concrete examination and discussion of the reasons why the proposed methdology is effective would yield a stronger paper. Intuitively, this action discretization seems to add the inferential bias that actions close to each other in continuous space are close to each other in value. This may be a generally reasonable assumption, but it would be nice if the authors could think on and discuss points such as these. Consequently, the limitations section could also more clearly outline the types of tasks for which this method may not be suitable.

While it seems reasonable that an advantage of this approach is that Q-learning can be used on the resulting discrete action space, this claim would be stronger if substantiated with experimental evidence. As the authors point out, it is possible to use the coarse-to-fine action discretization with other learning methods. It may be reasonable to evaluate actor-critic methods using the discretized action/policy space. As the authors note, action selection using e.g. upper confidence bound as in Monte Carlo Tree Search may be more efficient than greedily selecting the highest value action. Explicit policy learning may also yield improvements in action selection.

**Quality Of The Limitations Section:**

2

**Questions For Rebuttal:**

* When reporting performance metrics such as success rate, please report the total number of trials somewhere easily accessible
* If possible, it would be nice to see benchmarking in simulation of coarse-to-fine action discretization with other learning methods than Q-learning.

**Robotics Focus:**

4

**Summary Of Paper:**

This paper proposes to use Q-learning for continuous control by decomposing the action space into a hierarchical sequence of discrete actions. The key idea is that this discretization enables value learning on continuous actions. The authors evaluate this method in simulation and on a physical robot.

**Summary Of Recommendation:**

Overall, this paper presents a simple framework for decomposing continuous actions into a hierarchical sequence of low-dimensional discrete actions. The authors demonstrate the efficacy of Q-learning using a simple critic network on this action space on a range of simulated benchmarks and real-world tasks. The paper is clearly presented, and while there is no shortage of hierarchical learning algorithms, this method seems to be a clever but effective formulation, at least within the scope of tasks evaluated.

---

### Official Review · Reviewer_D3eS · 2024-07-17

**Originality:** 3
**Technical Quality:** 3
**Clarity Of Presentation:** 3
**Potential Impact:** 3
**Recommendation:** 3
**Confidence:** 3

**Review:**

This paper is well-written, and the proposed method is novel with a certain impact on the study of action discretization. The performance has been validated through several experiments and ablation studies. The comparison methods and the explanation of results could be more thorough.

### Strengths

- The paper is well-written and easy to read.
- The proposed method efficiently discretizes continuous actions hierarchically.
- The paper demonstrates superior performance over existing methods through extensive experiments on various tasks in both simulation and real-world robotic environments.
- The proposed method includes several techniques beyond action discretization, and the ablation studies validate the effectiveness of each technique, providing valuable insights for future research.

### Weaknesses

- The experiments do not include comparisons with other methods that discretize actions. Although the superiority over existing methods [24, 25, 26] is claimed, it is not empirically verified.

- The learning of the critic requires the calculation of losses for each level, and the integration of distributional RL methods increases computational costs.
- From Table 1 (c), it is evident that C51 significantly contributes to performance improvement, but the explanation for this is unclear. While it is suggested that C51 can suppress overestimation, C51 is not primarily designed for overestimation suppression.
- In the DMC tasks, there is not a significant improvement compared to SAC. In some tasks, it performs worse than SAC and DrQv2.

**Quality Of The Limitations Section:**

3

**Questions For Rebuttal:**

- How do you explain the significant performance improvement when combining C51 with the coarse-to-fine structure?
- For other questions, please see the weaknesses section.

**Robotics Focus:**

4

**Summary Of Paper:**

This paper addresses the instability of actor-critic methods by proposing a value-based algorithm with discretized actions. The proposed method discretizes actions at multiple levels in a coarse-to-fine manner, allowing for precise actions while maintaining high sample efficiency. Techniques such as distributional RL methods and exploration noise adjustment are also incorporated into the proposed method. Experiments were conducted on RLBench's sparse reward visual manipulation tasks and real-world environments, showing performance superior to existing methods like DrQ-v2. Additionally, ablation studies validated the effectiveness of each technique.

**Summary Of Recommendation:**

This paper is well-written and the performance has been validated through several experiments and ablation studies. Although the comparison methods and the explanation of results might be insufficient, the proposed method is novel and has a certain impact on the study of action discretization. Therefore, I recommend a weak acceptance.

---

### Official Review · Reviewer_4fBs · 2024-07-21
**Continuous Control with Coarse-to-fine Reinforcement Learning**

**Originality:** 3
**Technical Quality:** 3
**Clarity Of Presentation:** 3
**Potential Impact:** 3
**Recommendation:** 3
**Confidence:** 3

**Review:**

This paper provides an interesting approach to discretizing the continuous action space method called Coarse-to-fine Reinforcement Learning, which proposes a framework that trains RL agents to zoom into a continuous action space in a coarse-to-fine manner, enabling the use of stable, sample-efficient value-based RL algorithms for fine-grained continuous control tasks. The idea of the paper is to iteratively discretize the action space at multiple levels for each action dimension, using the Q value of the previous layer to further zoom-into the action at the current layer and find the best action by using argmax of the Q value.

Strengthen
 - the paper is well-written and carefully presented
 - the idea of the paper is interesting and could be considered for value-based RL methods with continuous action settings.
 - experiments results show that the proposed method sound and

Weakness
 - not sure if the proposed method ensures convergence

The idea is interesting and the paper could be interesting to the robotics community but I am wondering if the proposed method could ensure convergence because I think it is not sure that we can find the optional action by zoom-into a single dimension of an action and based on the Q value of the previous layer that only choose the interval with highest Q value. Of course, this approach could help save a lot of samples and reduce the search space for continuous action, but it could sometimes make the algorithm fail because optimality at each dimension interval does not mean the optimality of the Q value over all the action spaces.

Also, when we zoom in at each layer, we do not have Markovian properties of the Q value defined in the paper, so I am not sure if the estimated Q value is correct.

**Quality Of The Limitations Section:**

3

**Questions For Rebuttal:**

I think optimality at each dimension interval does not mean the optimality of the Q value over all the action spaces. So, it could be interesting if the authors have an analysis for the case when the algorithm fails?

Do we need the Markovian properties of the Q value defined in the paper for the algorithm to work?

**Robotics Focus:**

4

**Summary Of Paper:**

This paper introduces Coarse-to-Fine Reinforcement Learning, a method for discretizing continuous action spaces to enable stable, sample-efficient value-based RL algorithms for fine-grained control tasks, but concerns about convergence and Markovian properties requirements remain

**Summary Of Recommendation:**

The paper still need further clarification but the idea and results are interesting. I recommend weak accept.

---

### Author Rebuttal · Authors · 2024-08-08

Dear Reviewers,

We sincerely thank you for your overall positive comments and insightful comments to improve the manuscript. Here we attach the `cqn_rebuttal.zip` that contains 3 figure files that are included in our responses to the reviewers.
- `drqv2plus_rlbench_distributional.pdf` for response [A2] to Reviewer D3es
- `cqn_dmc_distributional.pdf` for response [A2] to Reviewer D3es
- `cqn_actor_critic.pdf` for response [A2] to Reviewer qKY2

---

### Decision · Program_Chairs · 2024-09-04

**Decision:**

Accept

**Comment:**

All reviewers agree that the paper is well written and that the ideas are interesting and have demonstrated merit. On the other hand, all of them have technical questions. A shared concern is that the reasons for observed performance improvements are unclear, and that more experimental evaluation should be provided to justify certain claims.

Such evaluations have been provided and questions have been answered to the reviewers' satisfaction. However, no updated draft has been provided. An acceptance would imply the expectation that the authors update the paper, integrating the additional results and clarifications as given in the rebuttal.

Particular strengths include
- that the paper is well-written and easy to read,
- demonstrated superior performance over existing methods through extensive experiments on various tasks in both simulation and real-world robotic environments,
- informative ablation studies.

Weaknesses include
- limited comparisons with other methods that discretize actions,
- mixed performance in certain cases,
- limited insight into what exactly enables and limits the strengths of the method.